# Caspase-mediated cleavage of IRE1 controls apoptotic cell commitment during endoplasmic reticulum stress

Anna Shemorry[1]*, Jonathan M Harnoss[1], Ofer Guttman[1], Scot A Marsters[1], László G Kőműves[2], David A Lawrence[1], Avi Ashkenazi[1]*

[1]Cancer Immunology, Genentech, South San Francisco, United States; [2]Department of Pathology, Genentech, South San Francisco, United States

**Abstract** Upon detecting endoplasmic reticulum (ER) stress, the unfolded protein response (UPR) orchestrates adaptive cellular changes to reestablish homeostasis. If stress resolution fails, the UPR commits the cell to apoptotic death. Here we show that in hematopoietic cells, including multiple myeloma (MM), lymphoma, and leukemia cell lines, ER stress leads to caspase-mediated cleavage of the key UPR sensor IRE1 within its cytoplasmic linker region, generating a stable IRE1 fragment comprising the ER-lumenal domain and transmembrane segment (LDTM). This cleavage uncouples the stress-sensing and signaling domains of IRE1, attenuating its activation upon ER perturbation. Surprisingly, LDTM exerts negative feedback over apoptotic signaling by inhibiting recruitment of the key proapoptotic protein BAX to mitochondria. Furthermore, ectopic LDTM expression enhances xenograft growth of MM tumors in mice. These results uncover an unexpected mechanism of cross-regulation between the apoptotic caspase machinery and the UPR, which has biologically significant consequences for cell survival under ER stress.

DOI: https://doi.org/10.7554/eLife.47084.001

*For correspondence:
shemorry.anna@gene.com (AS);
ashkenazi.avi@gene.com (AA)

## Introduction

The endoplasmic reticulum (ER) mediates three-dimensional folding of newly synthesized proteins that are destined for membrane insertion or extracellular secretion (*Walter and Ron, 2011*; *Hetz, 2012*; *Wang and Kaufman, 2016*). Excess demand for protein assembly in the ER causes accumulation of unfolded proteins – a condition known as ER stress. The unfolded protein response (UPR) is an intracellular sensing-signaling network that detects ER stress and orchestrates ER adaptation to reestablish cellular homeostasis. The UPR drives physical and biochemical expansion of the ER, while temporarily abating protein-translational load, and promoting disposal of unfolded proteins through ER-associated degradation (ERAD). In metazoan cells the UPR consists of three ER-transmembrane proteins: inositol-requiring enzyme 1 (IRE1), protein kinase R-like kinase (PERK), and activating transcription factor 6 (ATF6) (*Walter and Ron, 2011*; *Hetz, 2012*; *Wang and Kaufman, 2016*). These proteins directly or indirectly detect ER stress via their ER-lumenal and transmembrane domains; in response, they transduce signals to the cytosol and nucleus to promote cellular adaptation. The ER chaperone BiP/GRP78 plays an important role in keeping UPR activation in check (*Amin-Wetzel et al., 2017*; *Karagöz et al., 2017*). IRE1 possesses tandem cytoplasmic serine/threonine kinase and endoribonuclease (RNase) enzymatic modules, tethered to the ER membrane through an 80-amino-acid linker region (*Figure 1A*). Upon sensing ER stress, IRE1 forms homodimers, which perform trans-autophosphorylation of the kinase moiety, leading to RNase engagement. The RNase activates the transcription factor spliced X-box protein 1 (XBP1s) through nonconventional mRNA editing. In turn, XBP1s activates numerous genes that promote ER adaptation, ERAD, and cytoprotection (*Acosta-Alvear et al., 2007*). The RNase module also helps reduce ER-

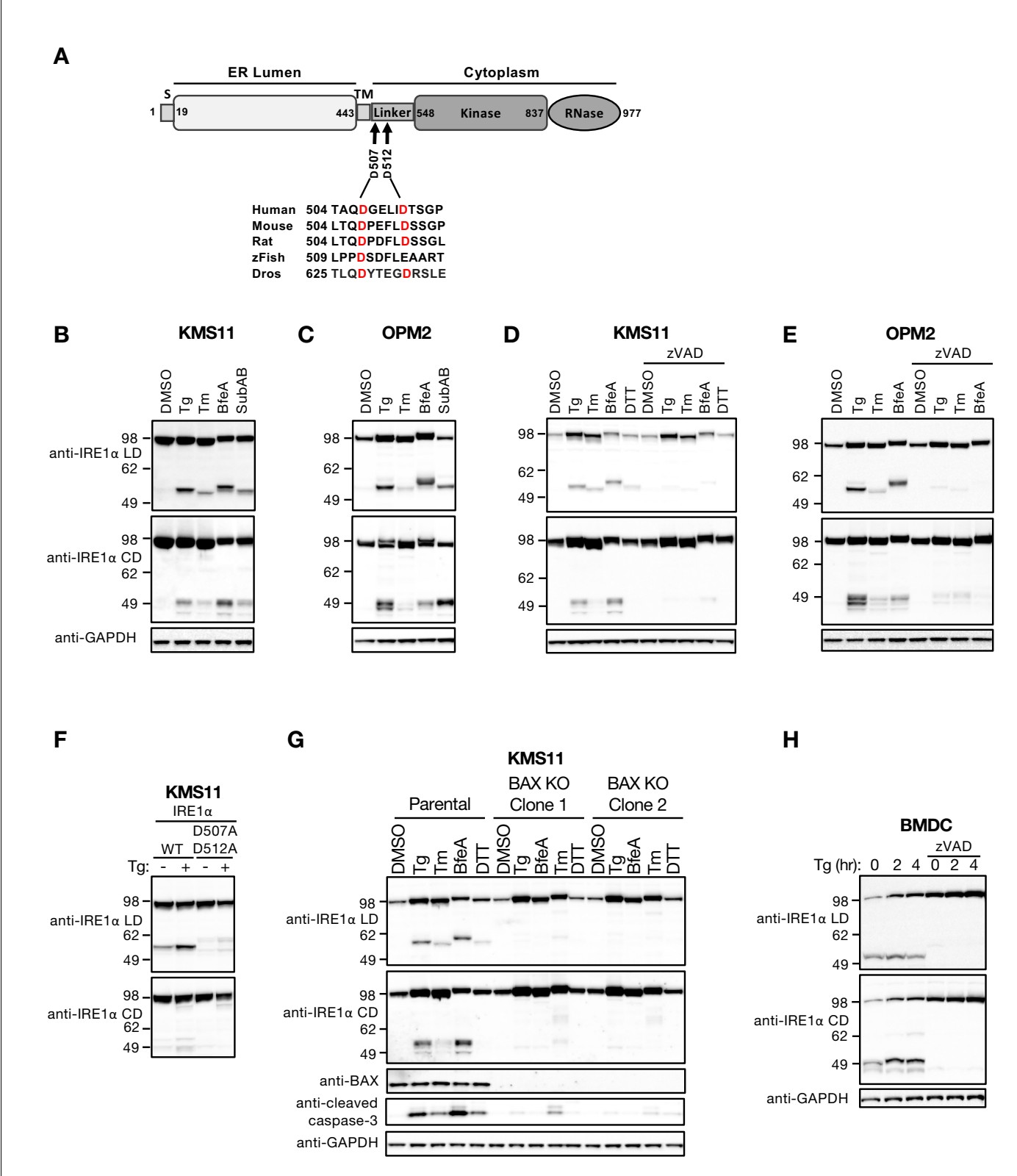

**Figure 1.** ER stress induces caspase-mediated cleavage of IRE1 in the cytoplasmic linker region. (**A**) Schematic representation of the human IRE1 protein and comparison of the amino acid sequences surrounding the predicted caspase cleavage sites in the linker region of IRE1 from different species (zFish, zebrafish; Dros, *Drosophila melanogaster*). (**B, C**) KMS11 (**B**) or OPM2 (**C**) cells were treated with 100 nM Tg, 5 µg/ml Tm, or 0.2 µg/ml BfeA for 16 hr, or 0.3 µg/ml SubAB for 3 hr. Cell lysates were analyzed by western blot (WB) with anti-IRE1α LD or anti-IRE1α CD antibody to detect the

*Figure 1 continued on next page*

*Figure 1 continued*

lumenal or cytoplasmic domains. (D, E) KMS11 (D) or OPM2 (E) cells were treated with 100 nM Tg, 5 µg/ml Tm, 0.2 µg/ml BfeA, or 1 mM DTT for 16 hr in the absence or presence of 20 µM zVAD. Samples were analyzed as in B for the presence of cleavage products. (F) A cDNA plasmid expressing either WT or doubly mutated IRE1 (D507A, D512A) was transiently transfected into KMS11 cells harboring CRISPR/Cas9-based IRE1 knockout. Cells were treated with either DMSO or 100 nM Tg for 16 hr and analyzed by WB as indicated. (G) Two independent KMS11 clones harboring CRISPR/Cas9-based BAX knockout were generated and validated for BAX deletion as compared to the parental cell line. Cells were treated with DMSO, 100 nM Tg, 0.5 µg/ml BfeA, 5 µg/ml Tm, or 1 mM DTT for 24 hr and analyzed by WB. (H) BMDCs were obtained from C57/BL6 mice and treated with 100 nM Tg in the absence or presence of 20 µM zVAD for the indicated times. Equal amounts of protein from cell lysates were analyzed by WB. B–H show representative results from at least three similar experiments. DMSO vehicle was used as control.
DOI: https://doi.org/10.7554/eLife.47084.002

The following figure supplement is available for figure 1:

**Figure supplement 1.** ER stress induces caspase-mediated cleavage of IRE1 in the cytoplasmic linker region.
DOI: https://doi.org/10.7554/eLife.47084.003

associated translational load through regulated IRE1-dependent mRNA decay (RIDD) (*Hollien et al., 2009*), which also can contribute to cytoprotection by regulating specific, functionally relevant genes (*Lu et al., 2014*; *Bae et al., 2019*). IRE1α (sometimes called ERN1) is the most evolutionarily conserved of the UPR sensors, displaying structural and functional homology in yeast, flies, worms, fish and primates. PERK harbors a similar lumenal domain and a cytoplasmic serine/threonine kinase module, but lacks an RNase moiety. In response to ER stress, PERK phosphorylates eukaryotic translation-initiation factor α (eIF2α) (*Walter and Ron, 2011*). This suppresses general translation, yet induces expression of activating transcription factor 4 (ATF4). In turn, ATF4 stimulates a number of genes that support ER adaptation, including one encoding the transcription factor C/EBP homologous protein (CHOP). The main function of the eukaryotic UPR is to adapt the ER to dynamic changes in demand for protein folding. However, the metazoan UPR performs an important additional function: it triggers apoptotic cell elimination in the event that ER-stress mitigation fails (*Tabas and Ron, 2011*). Teleologically, this mechanism probably evolved to limit the threat to the whole organism posed by potentially rogue cells with a severely damaged ER.

Apoptosis is a built-in cell-suicide program that is shared by most metazoans (*Kerr et al., 1972*; *Ellis and Horvitz, 1986*; *Hengartner and Horvitz, 1994*). The core intracellular apoptotic machinery entails a family of cysteine-dependent, aspartate-specific proteases, called caspases (*Thornberry and Lazebnik, 1998*; *Salvesen and Ashkenazi, 2011*). Two interconnected signaling cascades control caspase activation in response to severe cellular stress or damage: the intrinsic (or mitochondrial) pathway and the extrinsic (or death-receptor) pathway (*Martin and Green, 1995*; *Ashkenazi and Dixit, 1998*; *Danial and Korsmeyer, 2004*). These signals often converge on the proapoptotic BCL2 family protein BAX (and/or its relative BAK), which mediates mitochondrial outer membrane permeabilization (MOMP) through oligomerization and pore formation (*Youle and Strasser, 2008*). This releases cytochrome C from the mitochondrial intermembrane space into the cytosol. In the cytosol, cytochrome C helps nucleate the apoptosome complex, which sets off an enzymatic cascade involving the initiator protease caspase-9 and the executioner proteases caspase-3 and caspase-7 (*Slee et al., 1999*). It is believed that these caspases bring about the cell's apoptotic demise through 'death by a thousand cuts" (*Martin and Green, 1995*; *Taylor et al., 2008*). The PERK branch of the UPR plays a key role in driving apoptosis in the context of irresolvable ER stress, through mechanisms that involve ATF4 and CHOP and converge on BAX (*Lu et al., 2014*; *Tabas and Ron, 2011*; *Puthalakath et al., 2007*; *Han et al., 2013*). An additional proapoptotic signal involves the ER-resident BCL2 family protein BOK, which is kept at low levels by ERAD but accumulates when ERAD is diverted toward unfolded proteins (*Llambi et al., 2016*). In contrast to PERK, physiological levels of IRE1 suppress apoptosis activation, by mediating degradation of DR5 mRNA via RIDD during the early phase of the UPR (*Lu et al., 2014*). If ER stress persists, PERK-dependent attenuation of IRE1 via the phosphatase RPAP2 inhibits RIDD, allowing DR5 levels to rise and thereby driving apoptosis (*Chang et al., 2018*).

While the UPR exerts tight regulation over apoptotic cell commitment in the face of ER stress, it remains poorly studied whether the caspase machinery exerts reverse controls over the UPR. In mouse liver cells, BAX and BAK support activation of IRE1 by direct interaction with its cytoplasmic region (*Hetz et al., 2006*). Conversely, the ER-resident antiapoptotic protein BAX inhibitor-1 inhibits

IRE1 activation through cytoplasmic association (*Lisbona et al., 2009*). In addition, there is preliminary evidence that caspases may cleave PERK (*Shimbo et al., 2012*) and IRE1 (*Tang et al., 2018*); however, this has not been further investigated at the molecular and cellular levels. In the present study, we uncover a novel, unexpected mechanism of cross-regulation between apoptotic caspases and the UPR, which regulates cell survival during ER stress.

## Results

### ER stress promotes caspase-mediated cleavage of IRE1

Human IRE1$\alpha$ (herein IRE1) is a 977 amino-acid protein (*Figure 1A*), composed of an N-terminal ER-lumenal domain (LD), a single-pass transmembrane (TM) domain, and a C-terminal cytoplasmic domain (CD). The CD segment consists of three parts: a membrane-proximal 'linker' region, followed by a serine/threonine kinase domain, and an endoribonuclease (RNase) moiety. In exploring the biochemical fate of the IRE1 protein during ER stress, we obtained western blot (WB) evidence suggesting that full-length IRE1 undergoes proteolytic processing in cell lines derived from cancer patients with a B cell malignancy called multiple myeloma (MM) (see below). To follow up on these initial findings, we immunized mice with a purified recombinant human IRE1 LD protein and isolated a mouse IgG2a monoclonal antibody that selectively recognizes a specific epitope within the LD (anti-IRE1$\alpha$ LD). We then used this antibody, alongside a commercially available rabbit polyclonal antibody that specifically detects IRE1's RNase domain (anti-IRE1$\alpha$ CD), to further characterize the apparent processing of IRE1. We first interrogated two MM cell lines, KMS11 and OPM2, in which ER stress can be induced with classical pharmacological agents, i.e., thapsigargin (Tg), tunicamycin (Tm), brefeldin A (BfeA), and dithiothreitol (DTT), or the pathophysiological bacterial toxin subtilase AB5 (SubAB) (*Paton et al., 2006*). WB analysis with the anti-IRE1 LD antibody revealed not only the presence of the full-length ~105 kDa IRE1 protein, but also the emergence under ER stress of an additional ~55 kDa band (*Figure 1B and C*), indicating the formation of an N-terminal, LD-containing IRE1 fragment. Parallel analysis with the anti-IRE1$\alpha$ CD antibody also revealed the full-length IRE1 protein, as well as the formation in response to ER stress of one or more additional bands of ~50 kDa (*Figure 1B and C*), indicating the production of one or more C-terminal, CD-containing fragments. Some variation in gel mobility of the IRE1 species occurred; although further investigation is needed, this size variation may be due to differences in N-linked glycosylation (*Liu et al., 2002*) or other post-translational modifications of IRE1 under different ER stress conditions.

Unmitigated ER perturbation can lead to caspase-dependent apoptosis (*Walter and Ron, 2011*; *Tabas and Ron, 2011*). We therefore reasoned that caspase activation might underlie the apparent proteolytic cleavage of IRE1 during ER stress. Treatment of KMS11 or OPM2 cells with the pan-caspase inhibitor zVAD blocked generation of both the LD- and CD-containing IRE1 fragments upon ER stress (*Figure 1D and E*), indicating that this cleavage of IRE1 requires caspase activity. We first attempted to identify the processing site(s) by N-terminal sequencing through mass spectrometry, using either purified recombinant IRE1 proteins of various lengths or immunoprecipitated endogenous IRE1 polypeptide from ER-stressed cells; however, this approach proved unsuccessful (data not shown). Instead, we turned to the CaspDB database (http://caspdb.sanfordburnham.org/) to examine the IRE1 polypeptide sequence for potential caspase recognition sites. Although a number of sites were predicted, in light of the estimated molecular mass of the observed fragments and the sequence conservation of IRE1 between diverse species, we postulated that a major cleavage site(s) resides within the linker region, after aspartic acid 507 (0.879 probability), or aspartic acid 512 (0.921 probability), or both (*Figure 1A*). To test this prediction, we replaced these two aspartic acids—either individually or simultaneously—with alanine residues by site-directed mutagenesis. We then transfected each mutant or a wild type (WT) IRE1 control construct into KMS11 cells harboring CRISPR/Cas9-based knockout of endogenous IRE1 and tested for caspase-mediated cleavage. Although alanine substitution of either aspartic acid 507 or 512 alone did not prevent IRE1 processing in response to ER stress (*Figure 1—figure supplement 1A*), replacement of both residues blocked most of the cleavage (*Figure 1F* and *Figure 1—figure supplement 1B*). Some residual bands were detectable by the anti-IRE1$\alpha$ LD antibody even with the double mutation, suggesting that cleavage can shift to alternative, less efficient sites. A band of ~85 kDa was further detected by the anti-IRE1$\alpha$ CD but not the anti-IRE1$\alpha$ LD antibody, independent of mutation (*Figure 1F*),

suggesting the existence of another cleavage site that disrupts the LD epitope. However, given its location in the ER lumen, this site is unlikely to be targeted by caspases, which reside mainly in the cytoplasm (*Salvesen and Ashkenazi, 2011*). An additional band that ran beyond the 49 kDa marker could be detected by the anti-IRE1α CD antibody (*Figure 1B-E*). However, this band was absent in cells expressing the double mutant (*Figure 1F*), suggesting that it may be a secondary product of cleavage at 507 and 512. Regardless, although different caspase-susceptible sites within IRE1 may be cleaved under diverse stress conditions, our data maps two prominent, adjacent cleavage sites to aspartic acids 507 and 512, within the cytoplasmic linker region of IRE1. We therefore focused our investigation on further elucidating this event and its cellular consequences.

Executioner caspase activation in response to ER stress often involves the proapoptotic protein BAX (*Lu et al., 2014*; *Puthalakath et al., 2007*). To verify whether the processing of IRE1 under ER stress requires BAX, we disrupted the BAX gene in KMS11 cells via CRISPR/Cas9 technology. We obtained two independent BAX knockout clones, which produced no BAX protein as expected; in contrast to the WT cells, these clones showed substantially less generation of cleaved (activated) caspase-3 and lacked any detectable IRE1 fragments in response to several ER stressors (*Figure 1G*). Thus, caspases operating downstream to BAX in the apoptotic signaling cascade induced by ER stress in KMS11 cells mediate the proteolytic processing of IRE1. Consistent with this conclusion, purified caspase-3 and caspase-7 performed concentration-dependent cleavage of a purified recombinant protein comprising the linker, kinase and RNase domains of IRE1 (IRE1 LKR), with slightly more efficient processing by caspase-3 (*Figure 1—figure supplement 1C*).

To verify that the cleavage of IRE1 was not a peculiarity of KMS11 and OPM2 cells, we expanded our analysis to other cell types. We observed a similar pattern of caspase-mediated IRE1 processing in additional MM cell lines, as well as in several different types of lymphoma cells (*Figure 1—figure supplement 1D–F*). In contrast, a number of non-hematopoietic cancer cell lines examined, including NCI-H441 (non-small cell lung carcinoma), JHH-1 (hepatocellular carcinoma), and A2058 (melanoma), did not display appreciable IRE1 cleavage upon ER stress (data not shown).

Aspartic acids 507 and 512 are conserved between human, mouse, rat and fruit-fly IRE1, while position 512 is occupied by glutamic acid in zebrafish IRE1 (*Figure 1A*). Consistent with the homology between the human and mouse orthologs, ER stress induced caspase-mediated cleavage of mouse IRE1 in two types of murine lymphoma cells (*Figure 1—figure supplement 1D and F*). Similar processing occurred also in mouse primary bone marrow-derived dendritic cells (BMDC), which already exhibited some IRE1 cleavage at baseline (*Figure 1H*). Thus, the results obtained so far suggest that ER-stress-induced caspase-mediated IRE1 processing occurs more readily in hematopoietic cells, including both malignant and non-malignant types.

## The major fragments of IRE1 differ in their cellular disposition

The presence of prevalent caspase cleavage sites in the cytoplasmic linker region of IRE1 suggested that their hydrolysis splits the protein into an N-terminal fragment that contains both the LD and TM segments (LDTM) and a C-terminal cytoplasmic domain fragment(s) containing the kinase and RNase moieties (CD). Subcellular fractionation of OPM2 cells confirmed that the LDTM polypeptide was associated primarily with the membrane compartment, whereas the CD products were found mostly in the cytosolic fraction (*Figure 2A*). To investigate the cellular persistence of these products, we tracked their levels in cells undergoing Tg-induced ER stress over an 8 hr period. We performed these studies in the presence of cycloheximide (CHX) to inhibit further biosynthesis of IRE1 precursor. In both OPM2 and KMS11 cells, the LDTM fragment increased in abundance over 8 hr after Tg addition, while the CD fragment accumulated over the first 4 hr and then declined (*Figure 2B and C*, *Figure 2—figure supplement 1A and B*). Remarkably, the amount of full-length IRE1 protein substantially decreased over this period of ER stress, while caspase inhibition by zVAD blocked the drop, preserving most of the initial full-length IRE1 protein (*Figure 2B and C*). Thus, in the absence of de novo protein synthesis, caspases can cleave much of the available cellular pool of IRE1 in response to ER stress.

Upon sensing of unfolded proteins by IRE1, the LD directs activation of the cytoplasmic kinase and RNase modules (*Korennykh and Walter, 2012*). Cleavage within the linker region separates the lumenal and cytoplasmic moieties and therefore should disrupt IRE1 function. To test this prediction, we used zVAD, aiming to block IRE1 processing. OPM2 cells showed detectable XBP1s protein at baseline (*Figure 2—figure supplement 1C*), perhaps due to immunoglobulin overproduction by this

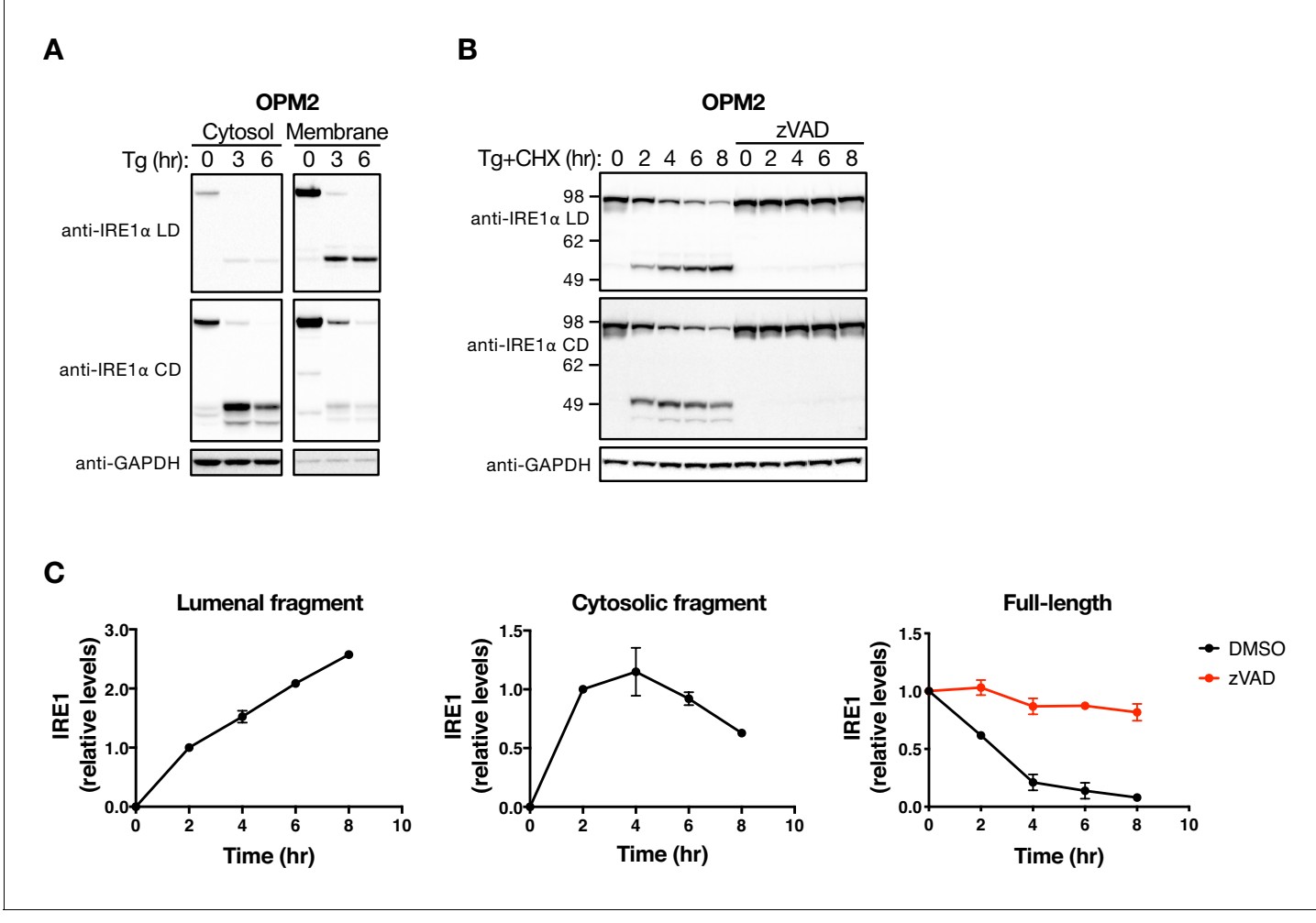

**Figure 2.** The LD and CD fragments of IRE1 differ in their cellular disposition. (A) OPM2 cells were treated with 100 nM Tg for the indicated time, subjected to subcellular fractionation, and the cytosol and membrane fractions were analyzed by WB. (B) OPM2 cells were treated with 100 nM Tg to induce ER stress as well as 10 µg/ml CHX to block protein synthesis, in the absence or presence of 20 µM zVAD to block caspase activity. After the indicated incubation time, cells were lysed and analyzed by WB. (C) Levels of the lumenal or cytoplasmic fragments or full-length IRE1 (anti-IRE1α LD) relative to GAPDH were quantitated using ImageJ and plotted as indicated. Data represent mean with standard deviation (SD) from two independent experiments. DMSO vehicle was used as control.

DOI: https://doi.org/10.7554/eLife.47084.004

The following figure supplement is available for figure 2:

**Figure supplement 1.** The LD and CD fragments of IRE1 differ in their cellular disposition.

DOI: https://doi.org/10.7554/eLife.47084.005

malignant plasma cell line. Treatment with zVAD before Tg addition indeed stabilized the full-length IRE1 protein and augmented IRE1 phosphorylation. Although XBP1s protein levels declined over time, likely due to the addition of CHX, zVAD treatment enhanced XBP1s production (*Figure 2—figure supplement 1C*), consistent with the stabilization of IRE1. BMDC showed no detectable XBP1s at baseline; nonetheless, their treatment with zVAD before Tg addition also stabilized the full-length IRE1 protein and augmented XBP1s production under ER stress (*Figure 2—figure supplement 1D*). Thus, caspase-mediated IRE1 cleavage dampens IRE1 activation by ER stress, suggesting that the apoptotic cascade feeds back onto IRE1 to reduce its cytoprotective activity.

## LDTM attenuates apoptotic caspase activation

The LDTM fragment of IRE1 persisted longer than the CD fragment. Therefore, we turned to investigate whether LDTM performs some specific cellular function. To examine this, we introduced (via

stable transfection) a cDNA plasmid construct encoding a Flag-epitope-tagged LDTM protein, driven by a cytomegalovirus promoter, into the KMS11 cell line. (Because this experiment was performed before we identified the precise cleavage sites, we first used a construct encoding amino acids 1–470, which ends within the linker region upstream to the 507 and 512 positions.) Whereas the parental cells displayed bands that matched the endogenous full-length IRE1 and its Tg-induced LDTM product, the transfected cells showed an additional band corresponding to the (shorter) ectopic LDTM protein, independent of ER stress (*Figure 3A*). Surprisingly, ectopic LDTM expression improved viability and diminished caspase-3/7 activation in KMS11 cells as compared to untransfected counterparts under Tg-induced ER stress (*Figure 3A and B*), indicating apoptosis attenuation. Removal of the C-terminal Flag tag from the cDNA construct did not alter LDTM's ability to attenuate caspase-3/7 activation in response to Tg (*Figure 3—figure supplement 1A*). Moreover, similar to the pooled transfected KMS11 cells, single-cell-derived clones overexpressing LDTM also displayed diminished caspase-3/7 activation as compared to parental cells upon ER stress induction by Tg or SubAB (*Figure 3C* and *Figure 3—figure supplement 1B*). To test an LDTM construct that more faithfully reflects the precise caspase cleavage site, we generated a cDNA plasmid encoding amino acids 1–507. Reassuringly, expression of this LDTM version also reduced caspase-3/7 activation in response to diverse ER stressors, including SubAB, Tg, Tm, and BfeA, in KMS11 or JJN3 cells (*Figure 3—figure supplement 1C, D, F and G*). These results demonstrate that LDTM inhibits apoptotic caspase activation upon ER stress. By extension, caspase-mediated cleavage of IRE1 during ER stress produces a membrane-tethered, lumenal product of IRE1 that in turn feeds back onto the apoptotic signaling cascade to inhibit further caspase activation. Consistent with this localization, immunofluorescence analysis showed co-staining of the transfected LDTM fragment with the ER membrane marker Calnexin in MDA-MB-231 cells (*Figure 3—figure supplement 1E*).

## LDTM operates independently of the UPR to attenuate apoptosis

To investigate mechanistically how LDTM inhibits caspase activation, we first asked whether its antiapoptotic activity requires the presence of full-length IRE1. To test this, we engineered KMS11 cells expressing an inducible shRNA construct that targets the 3' UTR of the IRE1 mRNA. We subsequently transfected these cells with a plasmid encoding Flag-tagged LDTM cDNA designed to resist the IRE1 shRNA. As expected, the parental cells (which harbor the shRNA but not the LDTM construct) expressed autologous full-length IRE1 and generated a corresponding LDTM product upon treatment with Tg (*Figure 3D*). Moreover, these parental cells showed complete depletion of the full-length IRE1 protein upon treatment with Doxycycline (DOX) to induce the shRNA, which blocked Tg-induced production of the endogenous LDTM fragment. As further expected, LDTM-transfected cells expressed the ectopic LDTM protein across all conditions. These cells similarly displayed complete depletion of the endogenous full-length IRE1 and its LDTM product upon DOX treatment. Importantly, cells expressing ectopic LDTM showed improved viability and diminished caspase-3/7 activation under Tg- or Tm-induced ER stress, regardless of DOX treatment (*Figure 3D and E*, *Figure 3—figure supplement 1F and G*). This data demonstrates that apoptosis attenuation by LDTM is equally effective in the presence or absence of endogenous IRE1. Thus, LDTM suppresses apoptotic caspase activation independently of full-length IRE1.

The UPR governs not only cellular adaptation but also apoptosis activation in response to protein misfolding. Accordingly, a conceivable mechanism that might mediate LDTM's antiapoptotic activity is that it binds to unfolded proteins in the ER lumen, in a manner similar to full-length IRE1 (*Gardner and Walter, 2011*), and sequesters them away from activating the ER-resident UPR sensors, thereby abating apoptotic signaling. However, consistent with its IRE1-independent function, ectopic LDTM expression did not alter IRE1 activity in response to ER stress, as measured by the levels of XBP1s mRNA (*Figure 4—figure supplement 1A*) or protein (*Figure 4—figure supplement 1B*), and by mRNA levels of the known RIDD target, DGAT2 (*Figure 4—figure supplement 1A*). In contrast to LDTM, ectopic expression of the CD fragment of IRE1 (LKR) did not affect caspase-3/7 activation by Tg or SubAB, nor did it impact LDTM's attenuation of caspase activation by these ER stressors (*Figure 4—figure supplement 1C and D*). Of note, LKR expression did not alter XBP1s levels (*Figure 4—figure supplement 1D*). Regardless, this data indicates that the LDTM product acts independently of the CD fragment to curb apoptosis signaling. Furthermore, LDTM expression did not inhibit ATF6 activation by ER stress, as gauged by levels of the key UPR chaperone BiP—a transcriptional target of ATF6 (*Walter and Ron, 2011*) (*Figure 4—figure supplement 1B*); nor did it

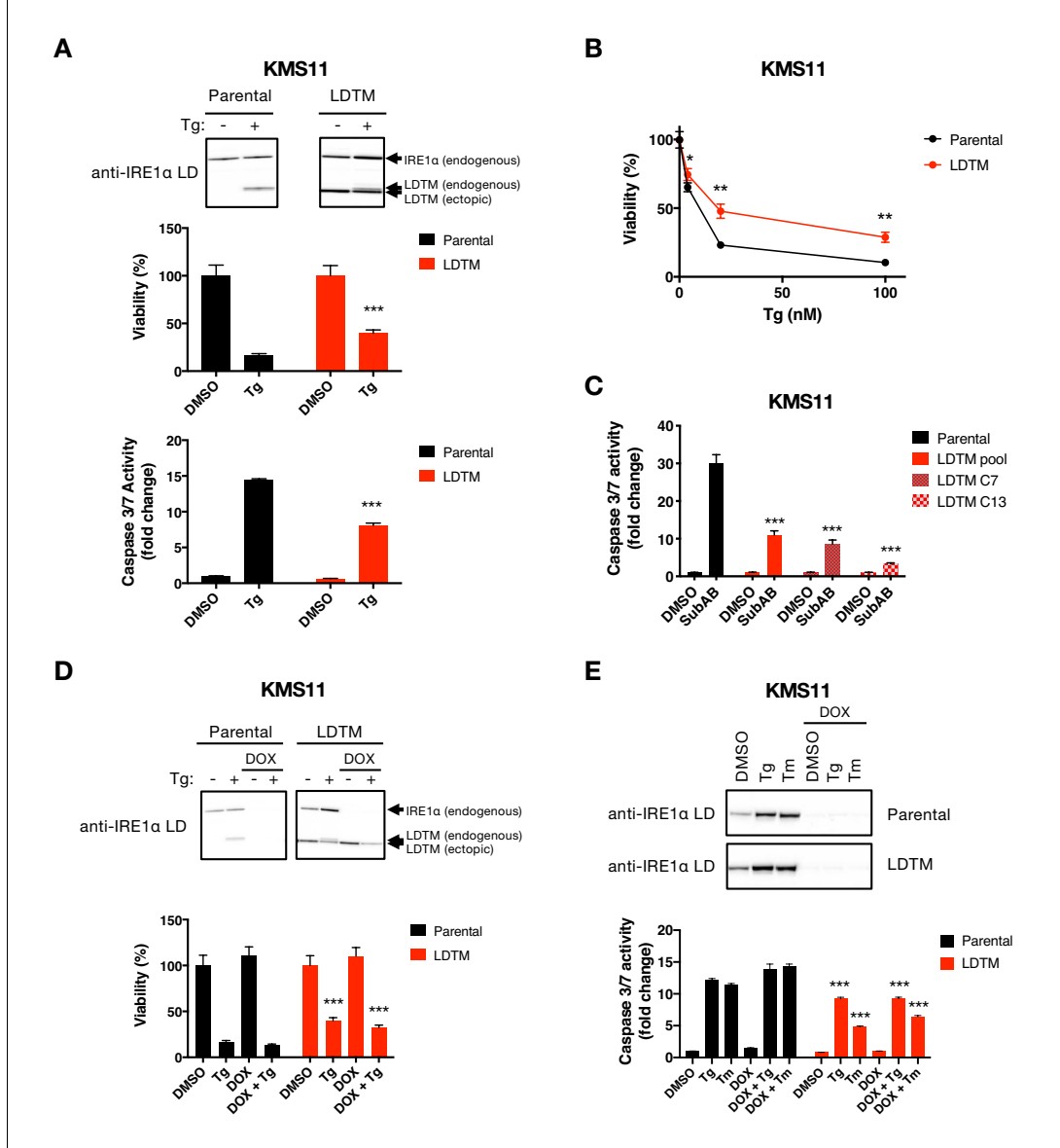

**Figure 3.** Ectopic expression of IRE1 LDTM attenuates apoptotic caspase activation independent of full-length IRE1. (**A**) KMS11 parental cells or cells stably expressing a cDNA plasmid encoding LDTM (1-470) driven by the CMV promoter were treated with DMSO or 100 nM Tg for 24 hr. Cell viability was measured using CellTiter-Glo normalized by the number of cells at seeding (middle panel). The percentage of viable cells is graphed as an average of three biological replicates. Equal amounts of protein from cell lysates were analyzed by WB (top panel) or Caspase-Glo 3/7 assay (bottom panel). WBs are representative of two or more experiments and the graph depicts mean ± SD of three technical replicates. (**B**) KMS11 cells as in **A** were treated with different concentrations of Tg for 24 hr and analyzed for viability. The percentage of viable cells is graphed as an average of three biological replicates ± SD. (**C**) KMS11 parental cells, LDTM expressing KMS11, or single cell clones derived from the LDTM transfected KMS11 pool (C7 or C13) were treated with 0.3 μg/ml SubAB for 3 hr. Equal amounts of protein from cell lysates were analyzed using the Caspase-Glo 3/7 assay. The graph depicts mean luminescence signal normalized to the control ± SD of three technical replicates. (**D**) KMS11 cells were stably transfected with a DOX-inducible shRNA plasmid targeting IRE1 (Parental). The cells were then stably transfected with a cDNA plasmid encoding LDTM (1-470) driven by the CMV promoter as in **A**. Parental and ectopic LDTM-expressing cells were treated for 3 days in the absence or presence of 1 μg/ml DOX to induce shRNA-mediated depletion of endogenous IRE1. Cells were then treated with DMSO or 100 nM Tg for 24 hr to induce ER stress and analyzed by WB (top panel) or CellTiterGlo assay (bottom panel). The percentage of viable cells is graphed as an average of three biological replicates ± SD. (**E**) KMS11 cells expressing DOX-inducible IRE1 shRNA were treated in the absence or presence of 1 μg/ml DOX and then subjected to ER-stress induction with 100 nM Tg or 5 μg/ml Tm for 24 hr. Cells were analyzed by WB (top panel) or Caspase-Glo 3/7 assay (bottom panel). The graph depicts mean luminescence signal normalized to DMSO ± SD of three technical replicates.

DOI: https://doi.org/10.7554/eLife.47084.006

The following figure supplement is available for figure 3:

*Figure 3 continued on next page*

*Figure 3 continued*

**Figure supplement 1.** Ectopic expression of IRE1 LDTM attenuates apoptotic caspase activation independent of full-length IRE1.

DOI: https://doi.org/10.7554/eLife.47084.007

decrease PERK activation, as judged by the levels of the PERK-regulated transcription factors ATF4 and CHOP, both of which are implicated in driving apoptosis downstream to PERK (*Tabas and Ron, 2011*; *Han et al., 2013*). LDTM also provided cell protection against SubAB—which induces ER stress by proteolytically cleaving BiP (*Lu et al., 2014*; *Paton et al., 2006*) (*Figure 3C*, *Figure 3—figure supplement 1C and D*). Thus, LDTM functions independently of the key UPR sensors, IRE1, ATF6 and PERK, as well as the major ER chaperone, BiP, to attenuate apoptotic caspase activation during ER stress.

## LDTM inhibits key mitochondrial apoptotic events

To further define how LDTM may attenuate caspase activation, we asked whether it acts upstream or downstream to specific mitochondrial events known to be crucial for an irreversible apoptotic cell commitment (*Martin and Green, 1995*; *Danial and Korsmeyer, 2004*). LDTM-transfected KMS11 clones displayed approximately 1.5-fold higher baseline levels of BAX as compared to parental controls (*Figure 4A*), perhaps reflecting a compensatory upregulation of BAX in response to the cytoprotection provided by LDTM. Nevertheless, ectopic LDTM expression attenuated the relative increase in mitochondrial BAX and the coordinated decrease in cytosolic BAX upon ER stress (*Figure 4A*). Consistent with its ability to decrease mitochondrial recruitment of BAX, LDTM expression also inhibited mitochondrial depolarization in cells undergoing ER stress in response to SubAB or Tg (*Figure 4B*). Furthermore, LDTM attenuated the uptake of calcium by mitochondria – a characteristic apoptotic event which was similarly curtailed by BAX KO (*Figure 4C*). LDTM also attenuated the Tg-induced drop in mitochondrial cytochrome C levels and the corresponding gain in cytosolic cytochrome C (*Figure 4D*). Moreover, LDTM attenuated Tg-induced activation of caspase-9, known to be triggered in the cytosol through cytochrome C-induced assembly of the apoptosome, as well as the activation of caspase-3/7 downstream (*Figure 4E*). Taken together, these results indicate that LDTM attenuates apoptotic caspase activation at the level of, or upstream to, BAX recruitment to mitochondria, thereby blocking the consequent steps of the apoptotic cascade; i.e., MOMP, mitochondrial calcium uptake, release of cytochrome C into the cytosol, and activation of caspase-9 and caspase-3/7. Further supporting the conclusion that LDTM regulates apoptosis via BAX, the BCL2 small molecule inhibitor ABT-199, which prevents BAX blockade by BCL2 (*Ashkenazi et al., 2017*), reversed LDTM's attenuation of mitochondrial calcium uptake and of caspase-3/7 activation in cells undergoing Tg-induced ER stress (*Figure 4—figure supplement 1E and F*).

## Ectopic LDTM expression enhances MM tumor progression

To establish whether the antiapoptotic effect of LDTM leads to a biologically significant consequence, we examined LDTM's impact on the growth of MM cells in vitro and in vivo. Ectopic LDTM expression augmented proliferation of KMS11 cells restricted on matrigel, as measured by confluence analysis over several days using an Incucyte S3 instrument (*Figure 5A*). To test whether this growth augmentation could still occur in the context of an in vivo microenvironment, which is expected to be more stringent, we subcutaneously xenografted the cells into SCID mice. Importantly, ectopic LDTM expression significantly enhanced the growth of KMS11 tumor xenografts in vivo (*Figure 5B*). Furthermore, while IRE1 depletion by shRNA inhibited KMS11 cell growth both in vitro and in vivo, in keeping with the cytoprotective role of this UPR sensor (*Harnoss et al., 2019*), expression of LDTM nevertheless accelerated growth even in the context of IRE1 knockdown (*Figure 5A and B*), consistent with LDTM's ability to improve cell viability independent of full-length IRE1. A single-cell-derived KMS11 clone expressing the same LDTM construct exhibited a similar enhancement in tumor growth (*Figure 5—figure supplement 1A*), as did a clone expressing the more precise 1–507 cleavage product (*Figure 5—figure supplement 1B*). Of note, KMS11 tumors constitutively produced an endogenous LDTM fragment, suggesting that cell stress in the tumor microenvironment drives caspase-mediated cleavage of IRE1. Taken together, these results suggest that the N-terminal product of caspase-mediated IRE1 cleavage – modeled here by ectopic LDTM –

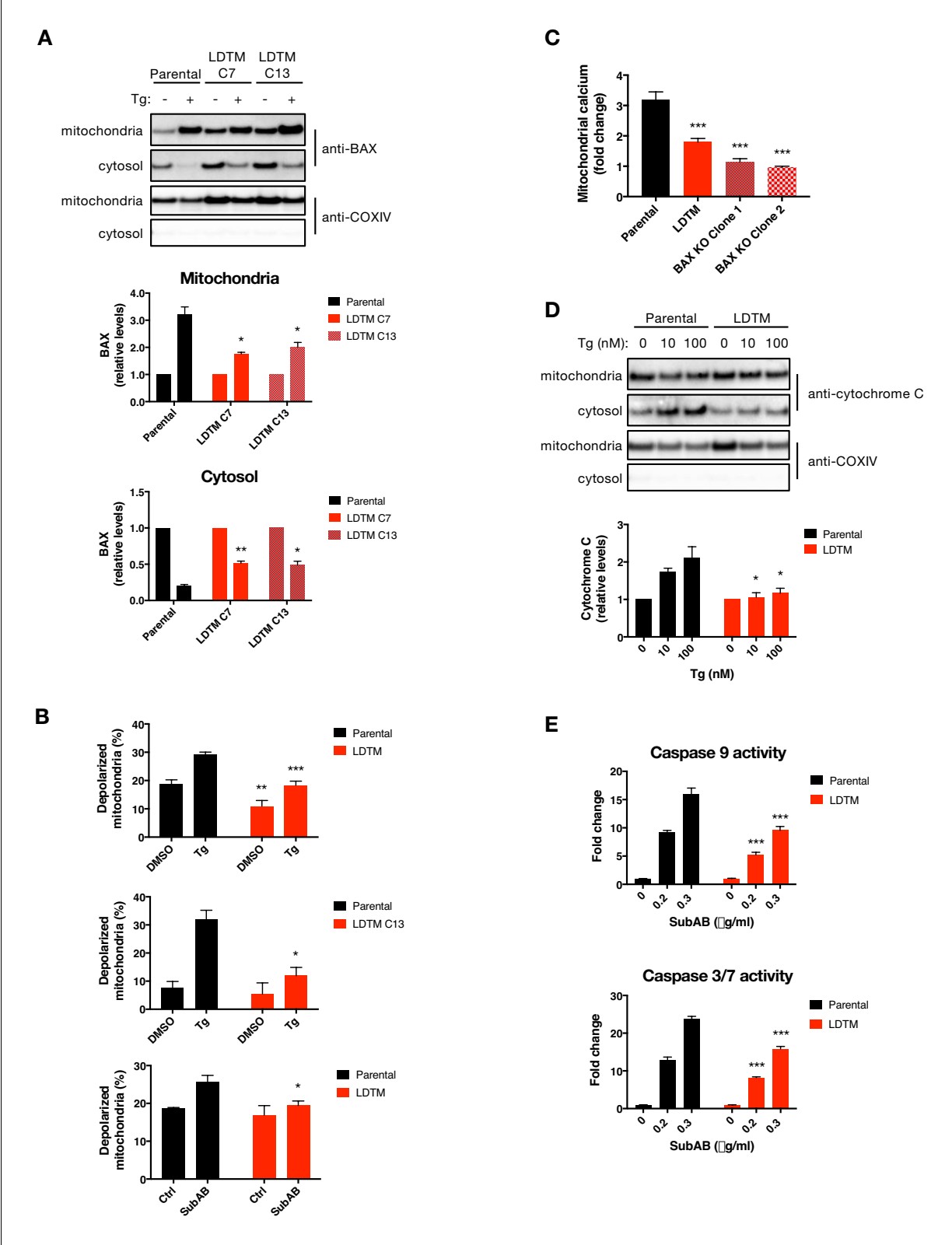

**Figure 4.** LDTM attenuates key mitochondrial apoptotic events. (**A**) Parental KMS11 cells or two clones expressing ectopic LDTM (1-470) were treated with DMSO or 100 nM Tg for 20 hr. Cells were differentially lysed to enrich for mitochondrial or cytoplasmic fractions and equal amounts of protein were analyzed by WB (top). Mitochondrial BAX levels were quantitated by ImageJ relative to the mitochondrial marker COXIV; cytosolic levels were similarly quantitated and graphed in relation to the corresponding DMSO controls. Data represent mean ± SD from two independent experiments. (**B**)
*Figure 4 continued on next page*

*Figure 4 continued*

Parental KMS11 cells or LDTM overexpressing cells, either a pool (top panel) or clone 13 (middle panel) were treated with 100 nM Tg for 20 hr. Similarly, parental cells and the LDTM overexpressing pool were treated with 0.3 μg/ml SubAB for 3 hr (bottom). Cells were subsequently incubated with 2 μM JC-1 dye for 30 min and analyzed for mitochondrial depolarization by FACS based on a fluorescence emission shift from red (~590 nm) to green (~529 nm). The average percentage of cells exhibiting mitochondrial depolarization ± SD from two or more biological replicates is graphed. (C) Parental KMS11 cells, LDTM overexpressing cells or two cell lines harboring a BAX deletion were treated with 100 nM Tg for 24 hr. Cells were incubated with the mitochondrial calcium dye Rhod-2 and then analyzed by FACS. Data represent the mean fold change in fluorescence ± SD as compared to DMSO treated cells from three or more biological replicates. (D) Parental KMS11 cells or cells expressing ectopic LDTM (1-470) were treated with DMSO or 10 or 100 nM Tg for 20 hr and differentially lysed to enrich for mitochondrial or cytoplasmic protein. Equal amounts of protein were analyzed by WB (top) and cytosolic amounts of cytochrome C were quantitated by ImageJ and graphed relative to the corresponding DMSO controls (bottom). Bar graphs represent mean ± SD from two independent experiments. (E) Parental KMS11 cells or cells expressing ectopic LDTM (1-470) were treated with 0.2 or 0.3 μg/ml SubAB for 3 hr and analyzed by Caspase-Glo 9 (top) or Caspase-Glo 3/7 (bottom) assay. Graphs depict mean ± SD of three technical replicates.
DOI: https://doi.org/10.7554/eLife.47084.008
The following figure supplement is available for figure 4:

**Figure supplement 1.** LDTM attenuates caspase activation without affecting the UPR and its function is reversed by BCL2 inhibition.
DOI: https://doi.org/10.7554/eLife.47084.009

exerts sufficient biological impact on KMS11 cells to enhance their fitness to proliferate under growth-limiting conditions in vitro and in vivo.

## Discussion

The primary function of the eukaryotic UPR is to help cells adapt to dynamic changes in demand for ER-mediated protein folding. The metazoan UPR has acquired an additional role, which may be equally as important: to eliminate cells that have sustained irreparable ER stress and as such pose a threat to the whole organism. The UPR performs this latter function by engaging the cell's apoptotic caspase machinery. However, this requires tight control, so that cells do not commit prematurely to an irreversible apoptotic fate. We know from earlier work that PERK acts through ATF4 and CHOP to drive apoptosis in response to excessive ER stress, that is, through Bim (*Puthalakath et al., 2007*), DR5, and/or DR4 (*Iurlaro et al., 2017*; *Dufour et al., 2017*), triggering apoptotic signals that converge on BAX. BOK further drives apoptosis when ERAD is diverted toward other ER client proteins (*Llambi et al., 2016*). In contrast to PERK, IRE1 opposes apoptotic signaling during the initial phase of the UPR by suppressing DR5 mRNA levels through RIDD (*Lu et al., 2014*). If stress persists, PERK activity attenuates IRE1 by driving its dephosphorylation through the RPAP2 phosphatase, thereby releasing the brake on DR5 to promote apoptosis during the terminal UPR (*Chang et al., 2018*). These mechanisms exemplify stringent UPR control over apoptosis activation. However, whether the cell's apoptotic caspase machinery feeds back onto the UPR to affect ultimate cell fate has been unclear.

Our present studies uncover a novel mechanism of reciprocal cross-regulation between the UPR and the apoptotic caspase cascade during ER stress (*Figure 5D*). We show that caspase activation in response to diverse ER-stress stimuli leads to proteolytic processing of IRE1. Caspases operating downstream to BAX cleave IRE1 at two adjacent sites within the cytoplasmic linker region, thus dividing the protein into two major products. Given that activated apoptotic caspases often overlap in selectivity toward intracellular substrates, we did not attempt in this study to identify which specific BAX-driven caspases are involved in the cellular cleavage of IRE1. Nevertheless, both caspase-3 and caspase-7 were capable of cleaving a purified recombinant IRE1 LKR protein. In cells, the N-terminal product of IRE1, consisting of the ER-lumenal domain, transmembrane segment, and some residual linker sequence (LDTM), remains membrane-anchored and persists for at least 8 hr after ER stress exertion. In contrast, the C-terminal product(s), which contains most of the cytoplasmic region, displays a more transient nature. By uncoupling the sensing and signaling domains of IRE1, caspase-mediated processing within the linker dampens activity of this key UPR sensor in response to ER stress.

Because the N-terminal fragment appeared more stable, we chose to focus on exploring its potential function. We found unexpectedly that upon overexpression, this product of IRE1 mediates significant negative feedback onto the apoptotic signaling cascade. In contrast, the more labile

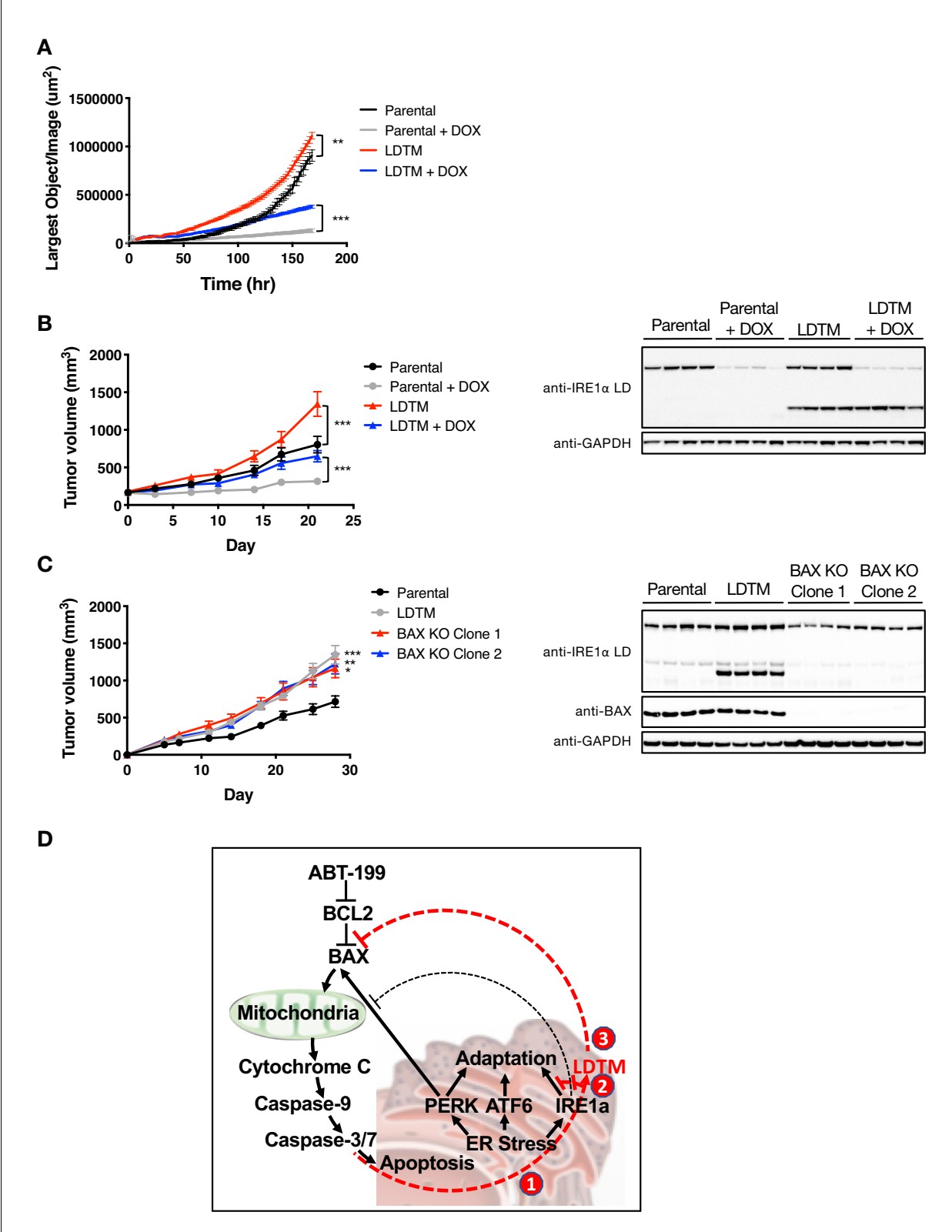

**Figure 5.** LDTM augments MM tumor progression. (**A**) Parental KMS11 cells or cells expressing ectopic LDTM (1-470) were plated on matrigel and growth was monitored by the changes in confluence using an IncuCyte S3 instrument over 7 days. (**B**) KMS11 parental cells expressing DOX-inducible IRE1 shRNA or the same cells transfected with a plasmid encoding CMV-driven LDTM (1-470) were injected subcutaneously into CB-17 SCID mice. When tumors reached ~150 mm³ in volume, mice were divided into groups (n = 9) and treated with sucrose or DOX via the drinking water and tumor

*Figure 5 continued on next page*

*Figure 5 continued*

growth was monitored over 21 days. Tumors were harvested and lysates were analyzed by WB. (C) KMS11 parental (n = 20), LDTM overexpressing cells (n = 10), or two independent KMS11 clones harboring CRISPR/Cas9-based BAX knockout (n = 10 each) were injected subcutaneously into CB-17 SCID mice. Tumor growth was monitored over 28 days. After which, tumors were harvested and lysates were analyzed by WB. (D) Schematic model illustrating previously known (black text and lines) and novel (red text and lines) cross-regulation between the UPR and the apoptotic cascade. ER-stress-induced apoptotic signaling leads to caspase-dependent cleavage of IRE1 (1). This separates the sensing and signaling domains of IRE1, which dampens IRE1's known XBP1s- and RIDD-mediated cytoprotective activities (2). Furthermore, it generates a fragment containing the lumenal domain and transmembrane segment (LDTM), which in turn suppresses further apoptotic signaling by attenuating BAX translocation to mitochondria (3) in a manner that can be reversed by the BCL2 inhibitor ABT-199.

DOI: https://doi.org/10.7554/eLife.47084.010

The following figure supplement is available for figure 5:

**Figure supplement 1.** LDTM augments MM tumor progression.

DOI: https://doi.org/10.7554/eLife.47084.011

C-terminal fragment did not affect key apoptotic indicators, nor did it functionally influence the N-terminal fragment. LDTM consistently restricted caspase activation in the face of ER stress. Furthermore, it acted without requiring full-length IRE1 or altering the engagement of PERK and ATF6, and it abated apoptosis even in response to SubAB, which causes ER stress by cleaving the central ER chaperone BiP. Together, these results indicate that LDTM operates independently of the UPR to inhibit further caspase activation. Additional mechanistic studies demonstrated that LDTM inhibits the recruitment of BAX from cytosol to mitochondria, thereby attenuating a number of critical mitochondrial and cytosolic events required for cell commitment to apoptosis. Precisely how LDTM suppresses BAX recruitment at the molecular level remains an open question. We have begun to explore this through proteomics approaches, seeking to identify specific LDTM interaction partners that might mediate its regulatory effect on the apoptotic cascade. The analysis so far has excluded the possibility that LDTM directly interacts with BAX (data not shown), in contrast to the earlier reported interaction of full-length IRE1 (via its cytoplasmic region) with BAX (*Hetz et al., 2006*). Of note, the BCL2 inhibitor ABT-199 overcame the cytoprotective effect of LDTM, thus reinforcing the importance of BAX regulation by caspase-dependent IRE1 cleavage.

The processing of IRE1 by caspases during ER stress seems paradoxically to have two opposite outcomes (*Figure 5D*): on the one hand it enhances apoptotic signaling by disrupting IRE1's activity, which is cytoprotective, while on the other hand it attenuates further caspase activation by restricting BAX translocation to mitochondria. A simple and plausible explanation for this apparent paradox is that the generation of LDTM through IRE1 cleavage provides a fail-safe checkpoint to ensure that cells with still remediable ER stress do not commit unnecessarily to apoptotic suicide. Only if ER stress exceeds a cell's capacity for mitigation does sufficient caspase activity develop to override this checkpoint and fully execute apoptotic elimination.

The cytoplasmic location of the prominent cleavage sites within IRE1 is consistent with the fact that caspase activity resides primarily in the cytosol. Furthermore, the linker region of IRE1 may be more accessible to caspases than other, perhaps more buried, cytoplasmic domains. This could explain why caspase recognition sites within the linker are favored over four additional theoretical consensus sites found within the kinase and RNase domains of human IRE1. Of interest, earlier work characterizing ectopically expressed IRE1 in COS7 cells showed that presenilin-1 cleaves IRE1 within the linker (*Niwa et al., 1999*), which suggests that this region of IRE1 can be targeted also by other types of proteases. IRE1 processing by apoptotic caspases occurred primarily in hematopoietic cell types, of both malignant and non-malignant origin. Because hematopoietic cells tend to express relatively high levels of IRE1 mRNA, it is possible that caspase-mediated processing requires abundant IRE1 protein: however, this warrants further investigation. Remarkably, in the absence of de novo IRE1 biosynthesis, most of the available cellular pool of IRE1 protein was subject to caspase-dependent cleavage in response to ER stress. This suggests that substantial amounts of LDTM can be generated within the cell, with significant consequences. Indeed, our biological studies showed that LDTM overexpression enhances restricted MM cell growth in vitro, as well as tumor progression in vivo. Thus, the production of LDTM upon caspase-mediated cleavage of IRE1 can significantly improve the fitness of cancer cells to survive and grow in stressful microenvironments. This conclusion has important potential implications for the role of IRE1 in hematopoietic malignancies such as

MM, because it suggests that IRE1 promotes tumor growth not only in its general role as a cytoprotective UPR mediator (*Harnoss et al., 2019*) but also in particular through its caspase-driven antia-poptotic LDTM product. Our studies open the door to further investigating caspase-mediated processing of IRE1 and other UPR sensors in response to proteotoxic as well as other types of cell stress in settings of health and disease.

# Materials and methods

**Key resources table**

| Reagent type (species) or resource | Designation | Source or reference | Identifiers | Additional information |
|---|---|---|---|---|
| Strain, strain background (mouse) | CB-17 SCID | Charles River Laboratories | | |
| Strain, strain background (mouse) | C57BL/6 | Charles River Laboratories | | |
| Cell line (human) | KMS11 | Genentech | RRID: CVCL_2989 | Multiple myeloma |
| Cell line (human) | OPM2 | Genentech | RRID: CVCL_1625 | Multiple myeloma |
| Cell line (human) | JJN3 | Genentech | RRID: CVCL_2078 | Multiple myeloma |
| Cell line (human) | RPMI-8226 | Genentech | RRID: CVCL_0014 | Multiple myeloma |
| Cell line (human) | LP-1 | Genentech | RRID: CVCL_0012 | Multiple myeloma |
| Cell line (human) | U266B1 | Genentech | RRID: CVCL_0566 | Multiple myeloma |
| Cell line (human) | Ramos | Genentech | RRID: CVCL_0597 | Burkitt's lymphoma |
| Cell line (human) | Raji | Genentech | RRID: CVCL_0511 | Burkitt's lymphoma |
| Cell line (human) | BJAB | Genentech | RRID: CVCL_5711 | Burkitt's lymphoma |
| Cell line (human) | Daudi | Genentech | RRID: CVCL_0008 | Burkitt's lymphoma |
| Cell line (human) | Maver-1 | Genentech | RRID: CVCL_1831 | Mantle Cell Lymphoma |
| Cell line (human) | Jurkat | Genentech | RRID: CVCL_0367 | Acute T-Cell Leukemia |
| Cell line (human) | MDA-MB-231 | Genentech | RRID: CVCL_0062 | Triple negative breast cancer |
| Cell line (mouse) | BW5147.3 | Genentech | RRID: CVCL_4135 | Thymic Lymphoma |
| Cell line (mouse) | ABE8.1/2 | Genentech | RRID: CVCL_3487 | Pre-B Cell Lymphoma |
| Cell line (human) | BAX knockout Clone one and Clone 2 | This paper | | Isolated clones of CRISPR/Cas9 deletion of BAX gene in KMS11 |
| Cell line (human) | IRE1 knockout KMS11 | Genentech | | *Harnoss et al., 2019* |
| Biological sample (mouse) | BMDC | This paper | | Isolated from the tibia and femur bones of C57BL/6 mice |

*Continued on next page*

*Continued*

| Reagent type (species) or resource | Designation | Source or reference | Identifiers | Additional information |
|---|---|---|---|---|
| Transfected construct (human) | LDTM (1-470) | This paper | | IRE1 aa1-470 with or without C-terminal Flag tag under CMV promoter |
| Transfected construct (human) | 1-507/1-507F | This paper | | IRE1 aa1-507 with or without C-terminal Flag tag under CMV promoter |
| Transfected construct (human) | IRE1 shRNA | Genentech | | *Harnoss et al., 2019* |
| Transfected construct (human) | IRE1 wt | This paper | | Full-length wild-type IRE1 expressed from CMV promoter |
| Transfected construct (human) | IRE1 D507A | This paper | | Asp at 507 was mutated to Ala |
| Transfected construct (human) | IRE1 D512A | This paper | | Asp at 512 was mutated to Ala |
| Transfected construct (human) | IRE1 D507A, D512A | This paper | | Both Asp at 507 and 512 were mutated to Ala |
| Transfected construct (human) | IRE1 LKR | This paper | | IRE1 aa468-977 with N-terminal His tag expressed from CMV promoter |
| Transfected construct (human) | BAX gRNA_1 | This paper | | GCGGTGATGGA CGGGTCCG |
| Transfected construct (human) | BAX gRNA_2 | This paper | | TTCATGATCTG CTCAGAGC |
| Antibody | GAPDH-HRP | Cell Signaling Technology | Cat. #: 2118 | (1:5000) |
| Antibody | XBP1s (rabbit monoclonal) | Genentech (*Chang et al., 2018*) | | (1:1000) |
| Antibody | IRE1α LD (mouse monoclonal, IgG2a) | This study | | (1:1000) |
| Antibody | IRE1α CD (rabbit monoclonal) | Cell Signaling Technology | Cat. #: 3294 | (1:1000) |
| Antibody | BAX (rabbit polyclonal) | Cell Signaling Technology | Cat. #: 2772 | (1:1000) |
| Antibody | Cytochrome-C (rabbit monoclonal) | Cell Signaling Technology | Cat. #: 11940 | (1:1000) |
| Antibody | COXIV (rabbit monoclonal) | Cell Signaling Technology | Cat. #: 4850 | (1:1000) |
| Antibody | ATF4 (rabbit monoclonal) | Cell Signaling Technology | Cat. #: 11815 | (1:1000) |
| Antibody | BiP (rabbit monoclonal) | Cell Signaling Technology | Cat. #: 3177 | (1:1000) |

*Continued on next page*

Continued

| Reagent type (species) or resource | Designation | Source or reference | Identifiers | Additional information |
|---|---|---|---|---|
| Antibody | Cleaved caspase-3 (rabbit monoclonal) | Cell Signaling Technology | Cat. #: 9664 | (1:1000) |
| Antibody | CHOP (mouse monoclonal IgG2a) | Cell Signaling Technology | Cat. #: 2895 | (1:1000) |
| Antibody | Flag (mouse monoclonal IgG1) | Sigma-Aldrich | Cat. #: F1804 | (1:1000) |
| Antibody | Calnexin (rabbit polyclonal) | Abcam | Cat. #: ab22595 | (1 µg/ml) |
| Antibody | Anti-rabbit IgG HRP | Jackson ImmunoResearch Laboratories | Cat. #: 711-035-152 | (1:10,000) |
| Antibody | Anti-mouse IgG2a HRP | SouthernBiotech | Cat. #: 1080–05 | (1:10,000) |
| Antibody | pIRE1α (rabbit monoclonal) | Genentech (*Chang et al., 2018*) | | (1:500) |
| Recombinant DNA reagent | pRK.TK.Neo-IRE1 WT | Genentech | | IRE1 wild-type cDNA in pRK.TK.Neo backbone |
| Recombinant DNA reagent | pRK.TK.Neo-IRE1 D507A | Genentech | | IRE1 cDNA with Asp 507 mutated to Ala in pRK.TK.Neo backbone |
| Recombinant DNA reagent | pRK.TK.Neo-IRE1 D512A | Genentech | | IRE1 cDNA with Asp 512 mutated to Ala in pRK.TK.Neo backbone |
| Recombinant DNA reagent | pRK.TK.Neo-IRE1 D507A, D512A | Genentech | | IRE1 cDNA with Asp 507 and 512 mutated to Ala in pRK.TK.Neo backbone |
| Recombinant DNA reagent | pRK.TK.Neo-IRE1 1–470-Flag | Genentech | | IRE1 cDNA aa1-470 with C-terminal Flag in pRK.TK.Neo backbone |
| Recombinant DNA reagent | pRK.TK.Neo-IRE1 1–470 | Genentech | | IRE1 cDNA aa1-470 in pRK.TK.Neo backbone |
| Recombinant DNA reagent | pRK.TK.Neo-IRE1 1–507-Flag | Genentech | | IRE1 cDNA aa1-507 with C-terminal Flag in pRK.TK.Neo backbone |
| Recombinant DNA reagent | pRK.TK.Neo-IRE1 1–507 | Genentech | | IRE1 cDNA aa1-507 in pRK.TK.Neo backbone |
| Recombinant DNA reagent | pcDNA3.1.Zeo-IRE1 6xHis 468–977 | Genentech | | IRE1 cDNA aa468-977 with N-terminal His tag in pcDNA3.1.Zeo backbone |
| Commercial assay or kit | Caspase-Glo 3/7 Assay | Promega | G8090 | |

*Continued*

| Reagent type (species) or resource | Designation | Source or reference | Identifiers | Additional information |
|---|---|---|---|---|
| Commercial assay or kit | Caspase-Glo 9 Assay | Promega | G8210 | |
| Commercial assay or kit | CellTiter-Glo Luminescent Cell Viability Assay | Promega | G7570 | |
| Commercial assay or kit | MitoProbe JC-1 Assay Kit for Flow Cytometry | ThermoFisher Scientific | M34152 | |
| Commercial assay or kit | Subcellular Protein Fractionation Kit for Cultured Cells | ThermoFisher Scientific | 78840 | |
| Commercial assay or kit | Mitochondria Isolation Kit for Cultured Cells | ThermoFisher Scientific | 89874 | |
| Recombinant protein | LKR | This paper | | Purified N-terminally His-tagged IRE1 aa468-977 |
| Recombinant protein | Caspase 3 | Enzo Life Sciences | ALX-201–059 | |
| Recombinant protein | Caspase 7 | BioVision | 1087 | |
| Chemical compound, drug | Rhod-2, AM, cell permeant | Invitrogen | R1244 | |
| Chemical compound, drug | Thapsigargin, Tg | Tocris | 1138 | |
| Chemical compound, drug | Tunicamycin, Tm | Tocris | 3516 | |
| Chemical compound, drug | Brefeldin A, BfeA | Tocris | 1231 | |
| Chemical compound, drug | DTT | ThermoFisher Scientific | R0861 | |
| Chemical compound, drug | Cycloheximide, CHX | Sigma-Aldrich | C4859 | |
| Chemical compound, drug | Z-VAD-FMK, zVAD | R and D Systems | FMK001 | |
| Chemical compound, drug | Doxycycline, DOX | Clontech | NC0424034 | |
| Chemical compound, drug | ABT-199 | Genentech | G00376771 | |
| Chemical compound, drug | Subtilase toxin AB5, SubAB | *Paton et al., 2006* | | |
| Software, algorithm | Prism 7 | GraphPad | | |

*Continued on next page*

*Continued*

| Reagent type (species) or resource | Designation | Source or reference | Identifiers | Additional information |
|---|---|---|---|---|
| Software, algorithm | ImageJ | NIH | | |
| Software, algorithm | FlowJo 10.4 | FlowJo, LLC | | |

## Cell culture and experimental reagents

All cell lines were cultured in RPMI-1640 medium supplemented with 10% fetal bovine serum, 2 mM glutamine and penicillin plus streptomycin. Tm, Tg, DTT and BfeA were purchased from Sigma and Tocris. zVAD-FMK was from R and D Systems. Caspase-Glo 3/7, Caspase-Glo 9, and CellTiter-Glo assay kits were from Promega. Antibodies to the RNase domain of IRE1$\alpha$, (anti-IRE1$\alpha$ CD), ATF4, GAPDH, BAX, cleaved Caspase 3, BiP, and CHOP were from Cell Signaling Technology. Antibodies to phospho-IRE1$\alpha$, XBP1s and IRE1$\alpha$ LD were generated at Genentech. Doxycycline was from Clontech. Geneticin selective antibiotic was from GIBCO.

## Cell lines

A full list of cell lines used in this study are shown in the Key Resources Table. All cell lines were obtained or generated from an internal repository maintained at Genentech.

### Cell line authentication/quality control

Short Tandem Repeat (STR) Profiling: STR profiles are determined for each line using the Promega PowerPlex 16 System. This is performed once and compared to external STR profiles of cell lines (when available) to determine cell line ancestry. Loci analyzed: Detection of sixteen loci (fifteen STR loci and Amelogenin for gender identification), including D3S1358, TH01, D21S11, D18S51, Penta E, D5S818, D13S317, D7S820, D16S539, CSF1PO, Penta D, AMEL, vWA, D8S1179 and TPOX.

### SNP fingerprinting

SNP profiles are performed each time new stocks are expanded for cryopreservation. Cell line identity is verified by high-throughput SNP profiling using Fluidigm multiplexed assays. SNPs were selected based on minor allele frequency and presence on commercial genotyping platforms. SNP profiles are compared to SNP calls from available internal and external data (when available) to determine or confirm ancestry. In cases where data is unavailable or cell line ancestry is questionable, DNA or cell lines are re-purchased to perform profiling to confirm cell line ancestry. SNPs analyzed: rs11746396, rs16928965, rs2172614, rs10050093, rs10828176, rs16888998,rs16999576, rs1912640, rs2355988, rs3125842, rs10018359, rs10410468, rs10834627, rs11083145, rs11100847, rs11638893, rs12537, rs1956898, rs2069492, rs10740186, rs12486048, rs13032222, rs1635191, rs17174920, rs2590442, rs2714679, rs2928432, rs2999156, rs10461909, rs11180435, rs1784232, rs3783412, rs10885378, rs1726254, rs2391691, rs3739422, rs10108245, rs1425916, rs1325922, rs1709795, rs1934395, rs2280916, rs2563263, rs10755578, rs1529192, rs2927899, rs2848745, rs10977980.

### Mycoplasma testing

All stocks are tested for mycoplasma prior to and after cells are cryopreserved.

Two methods are used to avoid false positive/negative results: Lonza Mycoalert and Stratagene Mycosensor.

Cell growth rates and morphology are also monitored for any batch-to-batch changes.

## Transfection with cDNA

Cell lines were transfected with cDNA using Lipofectamine 2000 (ThermoFisher Scientific) according to manufacturer's protocol. Cells were treated and analyzed by WB after 48 hr or stably selected with either G418 or Zeocin for 2 weeks.

## CRISPR/Cas9 knockout of BAX

The gRNAs were cloned into pUC57_AIO_CMV_Cas9_T2_GFP, enabling co-expression of each sgRNA, Cas9, and an GFP-based selection marker following transient transfection into target cells. Transfection was with Lipofectamine 3000 (Invitrogen) according to manufacturer's protocol. At 24 hr after transfection, cells were washed once in PBS and resuspended in PBS media containing 3% BSA Fraction V. The cell suspension was then filtered through a 35 mm membrane followed by immediate FACS sorting using the GFP+ selection marker. Single cell clones (n = 96) were plated and grown. Clones producing colonies were tested for proper BAX disruption by immunoblot.

## Monoclonal antibody generation

A recombinant protein encoding the lumenal domain (LD) of human IRE1$\alpha$ (amino acids 1–443) was generated via a baculovirus expression system in SF9 cells and purified to homogeneity using a TEV-protease cleavable His6 tag. Mice were immunized using standard protocols and monoclonal antibodies were screened by western blot against recombinant purified IRE1$\alpha$ lumenal and cytoplasmic domain proteins or lysates from MDA-MB-231 cells expressing wild type IRE1 or harboring CRISPR/CAS9 knockout of IRE1. A mouse IgG2a monoclonal antibody that specifically and selectively detected the human IRE1$\alpha$ LD (Lum017) was thus isolated and cloned.

## Western blot analysis

Cells were lysed in a buffer containing 1% Triton X-100, 150 mM NaCl, 50 mM Tris-HCl supplemented with protease and phosphatase inhibitors (ThermoFisher Scientific). Samples were cleared, analyzed by SDS-PAGE, electro-transferred, and membrane was blocked with 5% dried nonfat milk powder in PBST, washed and then reacted with antibody and analyzed using ECL reagent (GE Healthcare or Invitrogen). Typically, equal protein amounts, as measured by a BCA assay (Thermo-Fisher Scientific), were loaded onto each lane of the SDS-PAGE gel.

## Caspase cleavage assay

Recombinant IRE1 LKR (aa 468–977) was expressed using Baculovirus in insect cells and was purified with nickel resin using the His-tag at the N-terminus of the protein. 75 ng of purified LKR was incubated with either 0.05 or 0.5 units of active recombinant caspase 3 (Enzo Life Sciences) or caspase 7 (BioVision) for 2 hr at 37°C. Reactions were TCA precipitated, resuspended in SDS loading buffer and then analyzed by western blot.

## Caspase activity assays

Cells were lysed as described for western blot analysis. Equal protein amounts, as measured by a BCA assay (ThermoFisher Scientific), were diluted in PBS to a total volume 100 µl. An equal volume of Caspase-Glo buffer was added and luminescence was measured after 1 hr.

## Isolation and differentiation of bone marrow-derived dendritic cells (BMDC)

The tibia and femur bones of C57BL/6 mice were thoroughly flushed internally with PBS to extract bone marrow cells. Cells were cultured in RPMI-1640 medium, containing 10% fetal bovine serum, 50 U/ml streptomycin/penicillin, 50 mg/ml L-glutamine, and 50 mM β2-mercaptoethanol (Sigma-Aldrich). The medium was supplemented with 20 ng/ml granulocyte–macrophage colony-stimulating factor (Biolegend, San Diego, CA) and 10 ng/ml interleukin-4 (Biolegend) for 9 days, with growth media being replenished on days 3 and 6 of culture, as described previously (Fernandez et al., Nature Medicine, 1999). BMDCs were verified to be $\geq$90% CD11c$^+$ MHC class II$^{high}$ by flow cytometric analysis.

## Immunofluorescence

For immunostaining, cells cultured on Lab-TekII Chamber slides were washed three times in PBS, fixed for 20 min in 4% paraformaldehyde (EMS) at room temperature, washed, and permeabilized with 0.2% Triton X-100 for 10 min at room temperature. The slides were then blocked with 5% goat serum (Jackson ImmunoResearch) in 3% BSA/PBS for 30 min at room temperature. Flag (Sigma) or calnexin (Abcam) antibodies were diluted in 3% BSA/PBS and incubated with cells at 4°C overnight.

After three washes with PBS, cells were incubated with secondary antibodies conjugated to Alexa Fluor 488 or Alexa Fluor 647 (Jackson ImmunoResearch) for 1 hr at room temperature. Slides were mounted with ProLong Gold Antifade Mountant with DAPI (Invitrogen) and viewed with Leica SP5 inverted confocal microscope using a 100X objective.

### Mitochondrial calcium assay

Rhod-2, am cell permeant dye (ThermoFisher Scientific) was prepared according to manufacturer's protocol. Cells were incubated with 10 µM dye for 45 min. After trypsinization, cells were washed once with PBS and analyzed by FACS in the FL-2 channel using FACSCalibur (BD Biosciences) flow cytometer.

### Mitochondrial outer membrane depolarization (MOMP) assay

MOMP was analyzed by FACS using the MitoProbe JC-1 assay kit from Molecular Probes according to the manufacturer's protocol.

### Mitochondrial isolation

Mitochondrial isolation kit (ThermoFisher Scientific) was used to isolate mitochondrial fractions following the manufacturer's protocol.

### Subcutaneous xenograft tumor growth studies

All procedures were approved by and conformed to the guidelines and principles set by the Institutional Animal Care and Use Committee (IACUC) of Genentech and were carried out in an Association for the Assessment and Accreditation of Laboratory Animal Care (AAALAC)-accredited facility. For tumor growth studies, $10 \times 10^6$ KMS11 IRE1 sh8-9, LDTM pool, LDTM C13, 1–507 C5, BAX Knockout Clone 1 or Clone 2, respectively, were suspended in HBSS, admixed with 50% Matrigel (Corning) to a final volume of 100 µl, and injected subcutaneously in the right flank of 6 to 7 week old female CB-17 SCID mice. Tumor size and body weight were measured twice per week. Subcutaneous tumor volumes were measured in two dimensions (length and width) using Ultra Cal-IV calipers (model $54–10 - 111$; Fred V. Fowler Co.). The tumor volume was calculated with the following formula: tumor size ($mm^3$) = (longer measurement $\times$ shorter measurement$^2$) $\times$ 0.5. For IRE1 knockdown studies, tumors were monitored until they reached a mean tumor volume of approximately 150 $mm^3$, and then to induce knockdown of IRE1α randomized into the following treatment groups: (i) 5% sucrose water (provided in drinking water, changed weekly), or (ii) doxycycline (0.5 mg/ml, dissolved in 5% sucrose water, changed 3x/week). When mice reached endpoint criteria (see below) or after a 21 day treatment cycle, mice were euthanized by cervical dislocation and subcutaneous xenografts harvested for immunoblot analysis.

Animals in all studies were humanely euthanized according to the following criteria: clinical signs of persistent distress or pain, significant body-weight loss (>20%), tumor size exceeding 2500 $mm^3$, or when tumors ulcerated. Maximum tumor size permitted by the IACUC is 3000 $mm^3$ and in none of the experiments was this limit exceeded.

### Statistical analysis

Graphs depict the mean ± SEM or SD of triplicates. Statistical analysis was performed by Student's *t* test in Prism. A single asterisk indicates $p<0.05$ in comparison to the relevant paired value (e.g., in *Figure 2A*, $p<0.05$ for the comparison of Tg-treated LDTM-transfected cells versus Tg-treated parental cells). Similarly, two asterisks indicate $p<0.01$ and three asterisks indicate $p<0.001$.

## Acknowledgements

We thank Ariel Chen for generating cell lines, members of the Antibody engineering department for mAb generation, Peter Liu and Wendy Sandoval for mass spectrometry assistance, and the Ashkenazi lab for discussions.

# Additional information

## Competing interests

Anna Shemorry: Anna Shemorry is affiliated with Genentech Inc. The author has no other competing interests to declare. Jonathan M Harnoss: Jonathan M Harnoss is affiliated with Genentech Inc. The author has no other competing interests to declare. Ofer Guttman: Ofer Guttman is affiliated with Genentech Inc. The author has no other competing interests to declare. Scot A Marsters: Scot A Marsters is affiliated with Genentech Inc. The author has no other competing interests to declare. László G Kőműves: László G Kőműves is affiliated with Genentech Inc. The author has no other competing interests to declare. David A Lawrence: David A Lawrence is affiliated with Genentech Inc. The author has no other competing interests to declare. Avi Ashkenazi: Avi Ashkenazi is affiliated with Genentech Inc. The author has no other competing interests to declare.

## Funding

The authors declare that there was no funding for this work.

## Author contributions

Anna Shemorry, Conceptualization, Data curation, Formal analysis, Methodology, Writing—original draft, Writing—review and editing; Jonathan M Harnoss, Conceptualization, Data curation, Formal analysis, Methodology; Ofer Guttman, Data curation, Methodology; Scot A Marsters, Conceptualization, Data curation, Methodology; László G Kőműves, Data curation, Formal analysis, Methodology; David A Lawrence, Conceptualization; Avi Ashkenazi, Conceptualization, Supervision, Methodology, Writing—original draft, Writing—review and editing

## Author ORCIDs

Anna Shemorry https://orcid.org/0000-0002-5797-3116
Jonathan M Harnoss https://orcid.org/0000-0003-0885-1347
Avi Ashkenazi https://orcid.org/0000-0002-6890-4589

## Ethics

Animal experimentation: All procedures were approved by and conformed to the guidelines and principles set by the Institutional Animal Care and Use Committee (IACUC) of Genentech (protocol #16-3257) and were carried out in an Association for the Assessment and Accreditation of Laboratory Animal Care (AAALAC)-accredited facility.

## Decision letter and Author response

Decision letter https://doi.org/10.7554/eLife.47084.014
Author response https://doi.org/10.7554/eLife.47084.015

# Additional files

## Supplementary files

• Transparent reporting form
DOI: https://doi.org/10.7554/eLife.47084.012

## Data availability

All data generated or analysed during this study are included in the manuscript and supporting files.

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
