## [Decision Letter]

Thank you for submitting your article "Caspase-mediated cleavage of IRE1 controls apoptotic cell commitment during endoplasmic reticulum stress" for consideration by *eLife*. Your article has been reviewed by three peer reviewers, one of whom is a member of our Board of Reviewing Editors, and the evaluation has been overseen by Vivek Malhotra as the Senior Editor. The reviewers have opted to remain anonymous.

The reviewers have discussed the reviews with one another and the Reviewing Editor has drafted this decision to help you prepare a revised submission.

Summary:

The reviewers find that the paper describes a very interesting study of the mechanism of how IRE1 negatively regulates apoptosis during UPR stress. The authors follow up the recent discoveries on the mechanisms by which cells die in response to persistent ER stress. The work presents a novel and exciting finding by which caspase-driven processing of the ER stress sensor IRE1 leads to production of two proteolytic fragments, one of which is shorter-lived and pro-apoptotic and the other of which is longer-lived and – surprisingly – anti-apoptotic. Overexpression of the pro-survival fragment protects stressed cells from death, both normal and tumor cells. The effect seems to be limited to cells of hematopoietic origin for reasons that are unclear. Most of the data are of excellent quality and the findings greatly increase our knowledge of the mechanisms for the decision of living versus dying in cells encountering ER stress. This manuscript is appropriate for publication in *eLife* provided the issues are addressed.

Essential revisions:

1) The actual caspase required for IRE1 cleavage has not been identified. Even though the authors used z-VAD as a broad spectrum caspase inhibitor, it may have significant off-target effects. Thus, it is essential that the authors provide direct evidence that IRE1 is cleaved by caspases. The authors should either knockdown or knockout Caspase 3/7 in their hematopoietic cell lines and show that it abolishes the generation of LTDM. It is also possible to complement these experiments in vitro with incubation of purified Caspase 3 or 7 with purified IRE1 and IRE1 mutants lacking the cleavage sites to show the generation of the LTDM.

2) Provide mechanism on how LTDM may inhibit BAX from localizing to mitochondria. Sub-cellular localization analyses (not just cytosol vs. mitochondria) addressing how BAX pool is redistributed would be useful.

3) Both fragments of IRE1 should be expressed in cells to see if the LTDM exerts its effect even in the presence of the pro-apoptotic fragment.

4) Compare the effects of BAX-inactivation on KMS11 tumor growth with that of LTDM.

In addition, please tone down on claims of LTDM as most of it is from over expression studies and the physiological relevance is yet to be established.

---

## [Author Response]

Essential revisions:

*1) The actual caspase required for IRE1 cleavage has not been identified. Even though the authors used z-VAD as a broad spectrum caspase inhibitor, it may have significant off-target effects. Thus, it is essential that the authors provide direct evidence that IRE1 is cleaved by caspases. The authors should either knockdown or knockout Caspase 3/7 in their hematopoietic cell lines and show that it abolishes the generation of LTDM. It is also possible to complement these experiments* in vitro *with incubation of purified Caspase 3 or 7 with purified IRE1 and IRE1 mutants lacking the cleavage sites to show the generation of the LTDM.*

This is a valid comment. The involvement of caspases in IRE1 cleavage is also corroborated by the effect of BAX knockout, which prevented both caspase activation and IRE1 cleavage (Figure 1G). We attempted to disrupt caspase-3 and -7 by RNAi but were unsuccessful due to poor knockdown efficiency. Nevertheless, to examine more directly whether caspases can cleave IRE1, we generated a purified, recombinant IRE1 protein comprising the linker, kinase and RNase domains (LKR). Incubation of this protein with commercially available purified caspase-3 or -7 revealed its direct cleavage by these proteases (new Figure 1—figure supplement 1C). This demonstrates a direct capacity of executioner caspases to cleave IRE1.

2) Provide mechanism on how LTDM may inhibit BAX from localizing to mitochondria. Sub-cellular localization analyses (not just cytosol vs. mitochondria) addressing how BAX pool is redistributed would be useful.

We attempted immunofluorescence studies with anti-BAX and the mitochondrial marker TOM20; however, this proved difficult due to the small size and minimal ER-excluded cytoplasm in multiple myeloma cells. Instead, we now provide data showing that LDTM inhibits calcium uptake by mitochondria similar to BAX knockout (new Figure 4C). We further show that the BCL2 small molecule inhibitor ABT-199 overrides the effects of LDTM on mitochondrial calcium uptake and caspase-3/7 activation (new Figure 4—figure supplement 1E and 1F), confirming that the antiapoptotic function is exerted over BAX.

3) Both fragments of IRE1 should be expressed in cells to see if the LTDM exerts its effect even in the presence of the pro-apoptotic fragment.

As noted above, it appears there is a misunderstanding that the cytoplasmic fragment is pro-apoptotic--we did not propose this in the original manuscript. Regardless, we now performed the suggested experiment, which showed that ectopic expression of the cytoplasmic LKR fragment by itself did not alter caspase activation under ER stress; nor did this fragment affect the anti-apoptotic activity of the lumenal fragment upon co-transfection (new Figure 4—figure supplement 1C and 1D).

4) Compare the effects of BAX-inactivation on KMS11 tumor growth with that of LTDM.

We have performed an additional in vivo study to examine the effect of BAX inactivation and observed that BAX knockout and ectopic LDTM expression in KMS11 cells similarly augment in vivo tumor growth (new Figure 5C).

In addition, please tone down on claims of LTDM as most of it is from over expression studies and the physiological relevance is yet to be established.

Please note that the identification of LDTM as a stable product of caspase-mediated IRE1 cleavage is based primarily on the detection of endogenous IRE1 and its cleavage products. Furthermore, the endogenous IRE1 fragment accumulates during ER stress while full-length IRE1 isstabilized by caspase inhibition. Nevertheless, to acknowledge the limitation that the reviewer alluded to we added the term “overexpression” to the Discussion text.